# Global Variation in *Escherichia coli mcr-1* Genes and Plasmids from Animal and Human Genomes Following Colistin Usage Restrictions in Livestock

**DOI:** 10.3390/antibiotics13080759

**Published:** 2024-08-12

**Authors:** Biel Garcias, Mayra Alejandra Flores, Mercedes Fernández, William Monteith, Ben Pascoe, Samuel K. Sheppard, Marga Martín, Martí Cortey, Laila Darwich

**Affiliations:** 1Department Sanitat i Anatomia Animals, Veterinary Faculty, Universitat Autònoma de Barcelona (UAB), 08193 Cerdanyola del Vallès, Spain; 2Department of Biology, University of Oxford, South Parks Road, Oxford OX1 3RE, UK

**Keywords:** antimicrobial resistance, colistin, *mcr-1*, IS*Apl1*, promoter variants, fitness cost, *Escherichia coli*

## Abstract

Antimicrobial resistance (AMR) is a significant global health threat, with multidrug-resistant (MDR) bacterial clones becoming a major concern. Polymyxins, especially colistin, have reemerged as last-resort treatments for MDR Gram-negative infections. However, colistin use in livestock has spread mobile colistin resistance (*mcr*) genes, notably *mcr-1*, impacting human health. In consequence, its livestock use was banned in 2017, originating a natural experiment to study bacterial adaptation. The aim of this work was to analyse the changes in the *mcr-1* genetic background after colistin restriction across the world. This study analyses 3163 *Escherichia coli* genomes with the *mcr-1* gene from human and livestock hosts, mainly from Asia (*n* = 2621) and Europe (n = 359). Genetic characterisation identifies IncI2 (40.4%), IncX4 (26.7%), and multidrug-resistant IncHI2 (18.8%) as the most common plasmids carrying *mcr-1.* There were differences in plasmids between continents, with IncX4 (56.6%) being the most common in Europe, while IncI2 (44.8%) was predominant in Asia. Promoter variants related to reduced fitness costs and IS*Apl1* showed a distinct pattern of association that appears to be associated with adaptation to colistin restriction, which differed between continents. Thus, after the colistin ban, Europe saw a shift to specialised *mcr-1* plasmids as IncX4, while IS*Apl1* decreased in Asia due to changes in the prevalence of the distinct promoter variants. These analyses illustrate the evolution of *mcr-1* adaptation following colistin use restrictions and the need for region-specific strategies against AMR following colistin restrictions.

## 1. Introduction

Antimicrobial resistance (AMR) has become one of the greatest threats to global public health, responsible for an estimated 1.27 million annual deaths [1]. Emerging multidrug-resistant (MDR) bacterial clones are becoming more frequent, and the treatment options available for this type of infection are limited. This has resulted in the use of some antibiotics, such as polymyxins, that were previously excluded due to side effects [2].

Polymyxins, including colistin (polymyxin E), are antibiotics that disrupt bacterial membranes, causing cellular death [3]. Initially discovered in 1947 as a byproduct of *Paenibacillus polymyxa* subsp. *colistinus*, colistin’s use in human medicine was limited due to renal and neurological side effects [4,5]. It was primarily restricted to ophthalmic and topical applications [6]. However, the rise of MDR Gram-negative bacterial infections has revived colistin’s use as a last-resort treatment [2]. Colistin is also extensively used in livestock for therapeutic and prophylactic purposes [7]. Initially, the threat to human health from livestock use was underestimated, as resistance was believed to arise from chromosomal mutations [8], and transmission from livestock was considered rare. However, the discovery of mobile colistin resistance (*mcr*) genes in 2015 revealed the potential for gene transmission from livestock to humans [9].

Consequently, the agricultural use of colistin was heavily restricted in China and Europe in 2016 [10,11], and this has led to a decline in the use of colistin in livestock farming. Spain is a great example of this since colistin sales passed from 51.09 mg/PCU (livestock Population Correction Unit) in 2015 to 9 mg/PCU in 2017 and to the current 0.4 mg/PCU [12]. This has mitigated the spread of isolates carrying the *mcr-1* gene [13,14], which is expected given the high fitness costs associated with the carriage of the gene in *Enterobacteriaceae* [15]. However, recent evidence [16] points to the emergence of variants with reduced fitness costs while maintaining resistance rates, achieved by lowering gene expression. These findings suggest that restricting the use of colistin alone will not be sufficient to eliminate resistant strains, and additional control methods are needed.

Therefore, exploring alternative ways to mitigate the impact of the spread of *mcr-1* genes (and associated AMR) is necessary. Understanding the genetic background of *mcr-1* is important to identify different transmission routes. Initially mobilised from *Moraxella* spp. chromosomal sequences by the insertion sequence IS*Apl1* to *Enterobacteriaceae* strains [17,18], *mcr-1*-carrying isolates have lost the IS*Apl1* insertion sequence and relied on plasmids for transmission [19,20]. IncI2, IncX4, and IncHI2 plasmids most often carry *mcr-1* genes [19,21,22,23]. Furthermore, despite evidence of regional differences in *mcr-1* plasmid epidemiology [24], studies following recent restrictions on colistin use have been confined to isolates of Chinese origin [25]. However, these studies have typically only investigated isolates from human infections [19,21], and it is important to include isolates from agricultural livestock animals as a likely gene pool reservoir [26].

In this study, we compared plasmid epidemiology and the genetic background of *mcr-1* genes from different geographical locations and sources (humans and livestock), analysing the adaptations that followed colistin restriction. Using an archived public dataset of 3163 *mcr-1*-positive *E. coli* genomes from the National Center for Biotechnology Information (NCBI) Pathogen Detection platform (accessed on 26 February 2023), we characterise the genomic context of *mcr-1* genes.

## 2. Results

### 2.1. mcr-1-Positive E. coli Are Globally Distributed in Humans and Livestock

We compiled a dataset of 3163 *E. coli* genomes containing *mcr*-1 genes that were isolated from human infections and livestock hosts, with accompanying information on the country of origin and collection date. Most of the isolates were of human origin (58%, *n* = 1833), followed by those from poultry (22.1%, *n* = 698), swine (17.8%, *n* = 562) and cattle (2.2%, *n* = 70) (Appendix A). Geographically, most of the genomes came from Asia (82.9%, *n* = 2621), with most isolates collected in China (59.6%, *n* = 1885). Europe also contributed 11.3% of the samples (*n* = 359) (Appendix A). The temporal distribution of the genomes spans the years 2015 to 2019 (Appendix A), encompassing the period around the introduction of restrictions on colistin use.

The resistome of each isolate was characterised in silico, revealing an average of 13.6 AMR genes per isolate. The distribution of these genes displayed a bimodal pattern, with most genomes carrying around 14 AMR genes and isolates with just two AMR genes (Appendix A). Most isolates were categorised as MDR, showing putative resistance to three or more antimicrobial classes [27]. *E. coli* isolates carrying the *mcr*-1 gene were also typed according to their MLST genes and designated sequence types (STs) (Appendix A). The globally distributed ST10 was most frequently identified (9.1%, *n* = 295), followed by ST48 (3.6%, *n* = 115), ST101 (3.3%, *n* = 107) and ST156 (3%, *n* = 96) (Figure 1A). Isolates with similar STs can be grouped into clonal complexes (Appendix A), where CC10 was the most common one (31.6%, *n* = 591), followed by CC101 (10.3%, *n* = 192), CC23 (5.5%, *n* = 102) and CC156 (5.1%, *n* = 96) (Figure 1B). Finally, it was shown that STs were distributed among all the species, showing no host specialisation (Figure 1C).

### 2.2. IncI2 and IncX4 Are Specialized for mcr-1 While IncHI2 Is a MDR

In the majority of isolates, *mcr-1* genes were inferred to be located on plasmids (93.5%). This was similar for all studied hosts, with the greatest discrepancy in isolates sampled from swine, where 10% of *mcr-1* genes were identified on chromosomal contigs, compared to <5% in other hosts. Also, it is interesting to note that 10% of bovine isolates contained two plasmids carrying *mcr-1* (Appendix A).

For a thorough understanding of genetic context, plasmids containing the *mcr-1* gene were typed (Table 1). IncI2 (40.4%, *n* = 1231), IncX4 (26.7%, *n* = 813) and IncHI2 (18.8%, *n* = 574) were the most common plasmids, and the rest of the replicon types were present in a frequency under 5%. Differences in AMR gene carriage were observed between the distinct plasmid incompatibility groups. Specifically, there were two *mcr-1* carriage types. First, *mcr-1* presence in a large plasmid, such as IncHI2, that contains multiple other AMR genes (an average of 15 AMR genes). Second, the presence in plasmids specialised for *mcr-1* carriage, such as IncI2 or IncX4, which rarely contain other AMR genes.

### 2.3. Mcr-1 Promoter Variants Are Associated with Specific Plasmids and Insertion Elements

We screened isolates for the presence of IS*Apl1* insertion sequences and catalogued their promoter and plasmid types (Figure 2). The presence of IS*Apl1* insertion sequences differed between plasmid types. Its presence statistically varied among the distinct plasmids (*p* < 0.001), being relatively common in IncHI2 (248/574, 42.9%) and IncI2 plasmids (173/1231, 14.1%), and rare in IncX4 plasmids (13/800, 1.6%). Nearly half of the remaining plasmid types contained an IS*Apl1* insertion sequence (48.4%). Promoter variants also differed between each plasmid type (*p* < 0.001). Wildtype (*n* = 383) and PV3 (*n* = 280) promoter variants were most common in IncX4 plasmids. Two subtypes of IncI2 plasmids were observed. The first was strongly associated with the presence of the IS*Apl1* insertion sequence and two different *mcr-1* promoter variants. The second Incl2 subtype included multiple other *mcr-1* promoter variants, including PV1 (1.7%, *n* = 353), PV2 (2%, *n* = 51), PV4 (0%, *n* = 32), SDV1 (1.5%, *n* = 204) or SNP1 (0.9%, *n* = 210) and rarely contained IS*Apl1* insertion sequences. Plasmids containing IS*Apl1* insertion sequences often had the consensus (44.1%, *n* = 268), SDV2 (54.4%, *n* = 182) or SNP2 (75%, *n* = 4) *mcr-1* promoter variant. However, regarding the variants not associated with IS*Apl1*, the trend was softer than for IncI2. While the loss was clear for PV1 (2.9%, *n* = 35), the levels were intermediate for other variants stated to not be associated with IS*Apl1* in the case of IncI2 such as SDV1 (22.7%, *n* = 22) and SNP1 (18.2%, *n* = 33). Finally, the rest of the plasmids followed a similar path to IncHI2, showing that *mcr-1* specialisation could have an effect on the adaptation pathways.

### 2.4. IncI2 and IncX4 Plasmids Predominate in Asia and Europe, Respectively

We compared isolate genomes from Asia and Europe and identified differences in the prevalence of plasmid incompatibility groups between continents (Figure 3). In Asia, the most common plasmid type was IncI2 (1130/2524, 44.8%), and IncX4 plasmids (196/346, 56.6%) were most common in Europe. However, there was some difference in plasmid prevalence between countries. In Asia, greater variation in plasmid types was observed in Thailand, Laos and Vietnam. There was greater consistency in plasmid types identified in European countries, although IncHI2 was the most common plasmid type in France (28/51, 54.9%) instead of IncX4.

Colistin is still used in livestock farming in Laos and Vietnam, and plasmid types identified in these countries differed compared to the rest of the world. IncHI1 (13/72, 18.1%) was detected in Laos and IncP1 (16/61, 26.2%) and p0111 (11/61, 18.03%) in Vietnam. Thailand, and with a smaller sample Lebanon (42/68, 61.8%), demonstrated similar plasmid profiles to Europe, with a preponderance of IncX4 type plasmids (129/246, 52.4%). Despite small sample sizes from other regions of the world, plasmid type prevalence resembled Asian isolates; for example, in South America, IncI2 was the most common plasmid type (496%), with Brazil being similar to Lebanon and Thailand, with a greater proportion of IncX4 plasmids (30/37, 81.1%). Finally, few isolates from North America and Oceania were available to include in the study and few conclusions can be drawn.

Plasmid promoter variants also differed in prevalence between continents and countries (Appendix A). IncX4 plasmid types with the consensus variant were common in Europe (56.7%, *n* = 111), while the PV3 promoter variant was most prevalent in Asia (especially in China) IncX4 plasmids (48.1%, *n* = 270) (Appendix A). A new promoter variant (SNP4) was dominant in Eastern Europe (Czech Republic and Poland). Finally, in Thailand, Lebanon and Brazil, where IncX4 plasmids were the most common, the same association with the consensus variant was found, which makes them very close to Europe despite the geographic separation.

Greater heterogeneity was observed globally for IncI2 plasmids (Appendix A). The most common promoter variants were PV1 (28.7%, *n* = 327), the consensus variant (18.9%, *n* = 215), SNP1 (18.1%, *n* = 205) and SDV1 (17.3%, *n* = 195). The greatest heterogenicity was observed among Chinese genomes, which also had the largest sample size with 898 isolates carrying *mcr-1* in an IncI2 plasmid. Some promoter variants were specifically associated with geographic regions, such as PV1 in Russia, SDV2 in Pakistan, PV4 in South America and PV2 in Lebanon. In IncHI2 plasmids, the consensus variant was the most common promoter variant worldwide (Appendix A).

### 2.5. Different Genome Adaptations Occur Following Withdrawal of Colistin in Europe and Asia

Colistin was restricted for agricultural livestock prophylaxis by most countries in 2017. Comparing the frequency of plasmid types and promoter variants revealed different adaptations to colistin restriction between continents. In Europe, the adaptation was associated with a replacement of the MDR plasmid IncHI2 with the *mcr-1* specialised IncX4 and the consensus variant by SNP4 (Figure 4A,B). In contrast, for most plasmid types identified in Asia, promoter variants remained stable, with the exception being IncX4 plasmids, where the PV3 promoter variant has been replaced by the consensus variant, in contrast with a previous study focused on China [16]. Thus, Chinese isolates (*n* = 1885) were further studied, promoter stability was found (Appendix A), and it was concluded that the variability was due to the overrepresentation of Thai isolates after colistin withdrawal.

Among the most common adaptations among isolates from Asia, consistent with other reports [28], was the loss of IS*Apl1* insertion sequences in IncI2 and IncHI2 plasmids (Appendix A). This potentially leads to reduced bacterial fitness costs for plasmid carriage, facilitating the spread of *mcr-1*. While many IncI2 plasmids retained their IS*Apl1* sequences before and after control measures, IncHI2 plasmid architecture changed, equalising all promoter variant levels. The IS*Apl1* loss strategy differed; in IncI2, IS*Apl1* loss was related to the replacement of associated variants (described above) around 2014, before the colistin ban. For IncHI2, IS*Apl1* loss began after the colistin restriction but was not tied to a decline in associated variants, remaining stable over the years (Figure 4C,D and Appendix A).

## 3. Discussion

The colistin resistance gene *mcr-1* was discovered in China in 2015 [9], and most research on its genomic epidemiology has been conducted there [19,21,25]. However, *mcr-1* has since been detected worldwide [20,29,30,31]. Using the growing number of genomes available in the NCBI Pathogen Detection platform, this study aimed to analyse the genomic context of *mcr-1* across different hosts and regions globally following the restriction of colistin use.

Most sequences in our study came from Asia, but we also had a significant sample from Europe, allowing for meaningful comparisons and insights into the European situation. However, samples from North America and Oceania were scarce despite the availability of numerous E. coli genomes in the NCBI. This scarcity may be attributed to colistin never being registered for animal use in these regions [32], and metagenomic analyses indicate that the *mcr* gene present in these areas is *mcr-9*, which is not associated with colistin resistance [33]. Similarly, sample sizes from South America and Africa were insufficient to draw robust conclusions.

There was no evidence of host segregation in the distribution of clones. The niche overlap of *mcr-1* clones is consistent with a shared gene pool for AMR genes and emphasises the importance of including livestock in surveillance efforts. Although we had substantial samples from humans, swine, and chickens, fewer genomes were available from bovines, likely due to a lower prevalence of the *mcr-1* gene in these animals [34,35,36].

The *mcr-1* gene was predominantly found on plasmids. Chromosomal *mcr-1* is linked with higher resistance and fitness [16], but its frequency suggests either recent emergence or other limitations on proliferation. Globally, the most common plasmids were IncI2, IncX4, and IncHI2, consistent with previous studies [24,37]. In countries like Laos and Vietnam, where colistin use in livestock remains widespread [38,39], other plasmids such as IncP1, p0111, and IncHI1 were more predominant, indicating lower specialisation in the three main plasmids under colistin selection pressure.

The colistin ban has led to a decrease in IS*Apl1* [21,25] and the emergence of mutations in the mcr-1 promoter region [16], reducing its fitness cost. We observed a relationship between both phenomena. IS*Apl1* presence was rare in IncX4 plasmids, as previously described [19,40]. For IncI2, two trends were observed: promoter variants associated with the presence of IS*Apl1* and those without it. The most common variant, PV1, does not confer a fitness advantage on its own, explaining its low association with IS*Apl1*, which could otherwise lead to plasmid destruction [28]. SDV1, another common variant, is linked to the loss of IS*Apl1* in IncI2 plasmids and provides a fitness benefit. This is in contrast to its increased presence of IS*Apl1* in IncHI2 plasmids. Further studies are needed to fully understand the relationship between plasmid type, promoter variants, and IS*Apl1* presence to comprehend *mcr-1* adaptations after the colistin ban since it seems that this relationship could vary according to the distinct plasmids.

Although the colistin ban has significantly reduced *mcr-1* prevalence, it has not been eliminated [25,41,42,43,44]. Antimicrobial resistance (AMR) genes typically confer a fitness cost [45], and this is true for *mcr-1* [15]. Our findings reveal differing adaptation strategies in Asia and Europe post-ban. In Europe, adaptation appears straightforward, with a shift from MDR plasmids IncHI2 to more specialised plasmids like IncX4 [24] (Figure 4A), which confer a fitness advantage [46]. The consensus variant was dominant before the colistin ban, but SNP4 became predominant post-ban, being the only variant present in 2020 and 2021. In Asia, the predominant plasmid was IncI2, a specialised *mcr-1* carrier that differs from IncX4 in its association with IS*Apl1*, as was already stated by a previous study [19]. IS*Apl1*′s loss over time was linked to a replacement of promoter variants, suggesting a summation effort to reduce fitness cost. This relationship is difficult to understand since it is not clear how a single nucleotide mutation could be associated with the loss of an insertion sequence and requires further investigation since other factors, such as an increase in plasmid transmission, could be implicated [47]. Nevertheless, it seems that the *mcr-1* adaptation strategy to cessation of colistin use was quite different in Asia and Europe.

Despite being the largest dataset used to study global *mcr-1* genomic context and evolution, this study has limitations. It is based on genomes from a public database (NCBI), which is not a stratified sampling, and conclusions cannot be drawn from the absence of *mcr-1*-positive samples in some regions due to unequal sampling. Nonetheless, it is important to make global comparisons, given the lack of comparative studies outside of China. Furthermore, the colistin ‘ban’ has not been applied evenly between countries, from complete prohibition to cessation of prophylactic and growth-promoting but continued therapeutic use under strict conditions. By highlighting region-specific *mcr-1* adaptations, our study provides a basis for future surveillance and informed interventions.

## 4. Materials and Methods

### 4.1. Isolate Genomes

The Microbial Browser for Identification of Genetic and Genomic Elements (MicroBIGG-E) provides a platform to search for genomes containing specific AMR genetic determinants present within the NCBI Pathogen Detection Database (26 February 2023). We conducted a search of all *E. coli* assemblies from human, swine, chicken and bovine sources that contained the *mcr-1* gene. Genome assemblies with available information regarding the country of origin and the year of collection were downloaded, resulting in a subset of 3185 *E. coli* assemblies. Genomes uploaded to the NCBI database theoretically have already passed a quality control (QC) process. However, to ensure the quality of our assemblies, we applied an additional QC step and discarded those assemblies with N90 < 1000 and those with lengths greater or less than 3 standard deviations from the mean size, resulting in a final dataset of 3163 assemblies. This information, together with the results of the analysis and the available metadata, is presented in Appendix A.

### 4.2. Multilocus Sequence Typing and AMR Genes General Detection

*E. coli* genomes were assigned to Sequence Types (ST) with PubMLST (https://pubmlst.org/) (accessed on 23 November 2023) [48], using the Achtman scheme [49]. eBurst [50] was used to assign ST to Clonal Complexes and to generate a minimum spanning tree. The presence of AMR genes across the whole genome was determined using the Resfinder database [51]. A positive hit was considered when a gene had >80% nucleotide identity over >80% of the sequence length. This allowed us to assess the general AMR levels, not limited to the *mcr-1* gene.

### 4.3. Genomic Context of mcr-1 Gene

To determine the genomic localisation of the *mcr*-1 gene, genome contigs were classified as either plasmid or chromosomal using the MOB-Recon tool from MOB-Suite [52], which employs a BLAST and bash approach to reconstruct plasmid sequences. Chromosomal contigs were scanned against the Resfinder database, and if *mcr*-1 was detected, it was classified as chromosomal. For contigs of plasmid origin, initial screening was conducted with the PlasmidFinder database [53], which identifies distinct plasmid incompatibility groups (Inc groups). These contigs were subsequently rescanned against the Resfinder database. If *mcr*-1 was detected within a plasmid, it was classified as plasmid-borne. To understand the plasmid’s role in AMR transmission, other AMR genes present on the same plasmid were identified. If *mcr-1* was found in two plasmids, both plasmid types were recorded to reflect their presence in the dataset. Finally, if the gene was encountered in the same isolate into the chromosome and a plasmid, the location was recorded as “Plasmid/chromosome”.

ISEScan [54] was used to detect insertion sequences. The number of IS*Apl1* (IS*30* family) copies was also identified, although, for clarity of the study, only the presence or absence of IS*Apl1* was reported.

Promoter variants, as identified by a previous study [16], were detected using blastn. These promoter variants are based on a *mcr-1* genomic comprehensive study [20], which identified polymorphism hotspots upstream of the *mcr-1* gene. Briefly, Ogunalana and colleagues tested the fitness cost of the wildtype promoter variant (that is, consensus variants), and we compared it with eight other different variants that were located in regions predicted to have a translation or expression effect. These zones were the Shine–Dalgarno (SD) sequence (the ribosomal binding site) and regions predicted to encode the RNA polymerase binding site (−10 and −35 boxes). Thus, nine variants were tested: consensus, four mutations in the RNA polymerase binding site (PV1, PV2, PV3 and PV4) and four mutations in the SD site (SDV1, SDV2, SDV3 and SDV4). Apart from this, eight more variants less frequent were detected in the Ogunalana’s dataset that were located in regions, theoretically, not implicated in the expression nor the fitness cost of *mcr-1* and were named SNP1, SNP2, SNP3, SNP4, SNP5, SNP6, SNP7 and SNP8. A positive hit was considered when identity and coverage were 100%. Sequences without a positive hit were manually inspected, and any new variants found were documented. We detected 21 promoter variants in addition to 16 previously identified variants (frequencies summarised in Appendix A). However, none of the newly described variants had a prevalence higher than 0.5%. To facilitate the visualisation of findings, the ten most common variants were considered, and those remaining were considered as “Others”.

### 4.4. Data Analysis and Visualization

Data analysis was performed with R software (version 4.2) [55], and graphs were built with ggplot2 (version 3.5.1) [56]. Statistics were used to detect differences between groups using the chi-square test. Maps were created using Microreact (https://microreact.org/) [57]. Data and maps can be consulted here: (https://microreact.org/project/eDrcgfHvuxVSRkbBK1Qx1b-mapamcr1).

## 5. Conclusions

In conclusion, our study highlights the need for region-specific analyses of *mcr-1* adaptations and underscores that findings from Asia should not be generalised globally. We mapped promoter variants associated with reduced fitness costs worldwide and identified trends like the potential SNP4 emergence in Europe and the IS*Apl1* loss in Asia due to the replacement of the promoter variants. These findings warrant further investigation to fully understand *mcr-1*′s adaptation mechanisms.

## Figures and Tables

**Figure 1 antibiotics-13-00759-f001:**
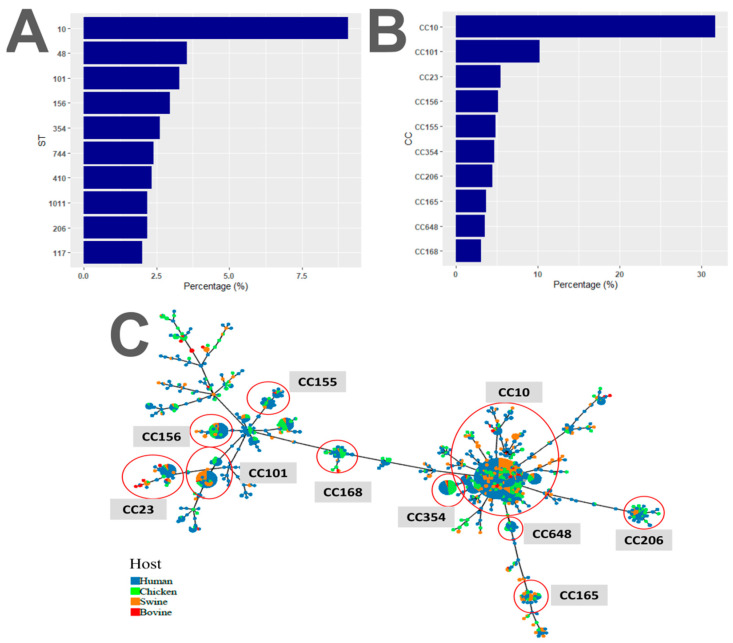
Population structure of genomes inferred to contain the *mcr-1* gene shows no host specialisation. (**A**,**B**) Bar plots showing the ten most common sequence types (ST) (**A**) and clonal complexes (CC) (**B**). (**C**) Minimum spanning tree showing ST distribution between hosts. The ten most common CCs are highlighted with a red circle. Host origin: human (in blue), chicken (in green), swine (in orange) and bovine (in red).

**Figure 2 antibiotics-13-00759-f002:**
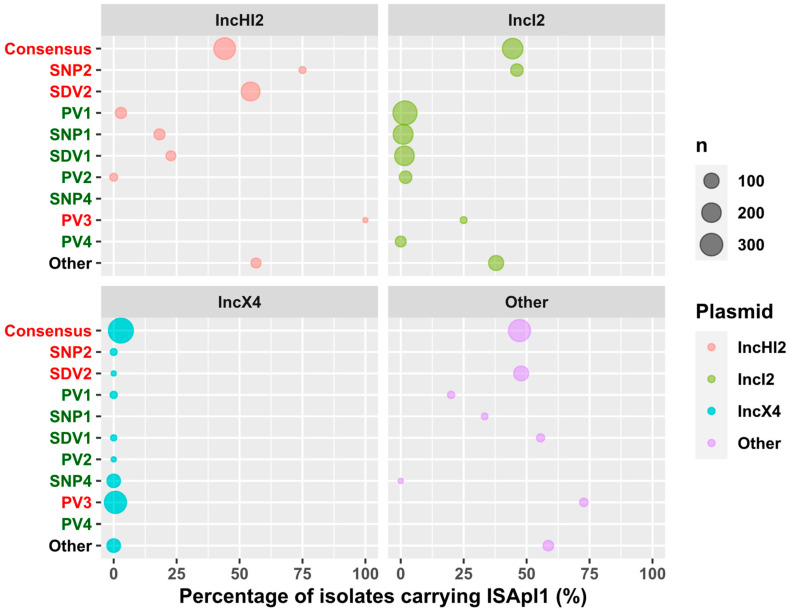
Association of the distinct plasmid types with promoter variants and inferred IS*Apl1* carriage. Strongly IS*Apl1*-associated variants (more than 40%) are highlighted in red, while those low associated (less than 5%) are in green. The size of the circle is directly proportional to the number of isolates of each category.

**Figure 3 antibiotics-13-00759-f003:**
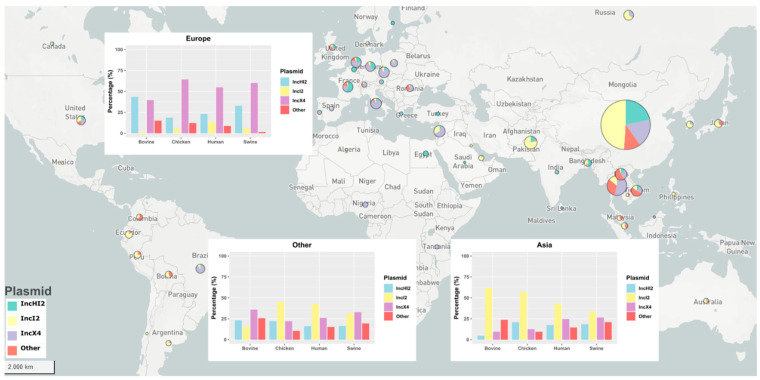
Global map of frequency of plasmid incompatibility (Inc) groups. Bar plots show the frequency of plasmid Inc groups of the different continents split by the isolate host source. Pie chart diameter is proportional to genome number per country. Colours represent IncHI2 (blue), IncI2 (yellow), IncX4 (purple), and other plasmids (red).

**Figure 4 antibiotics-13-00759-f004:**
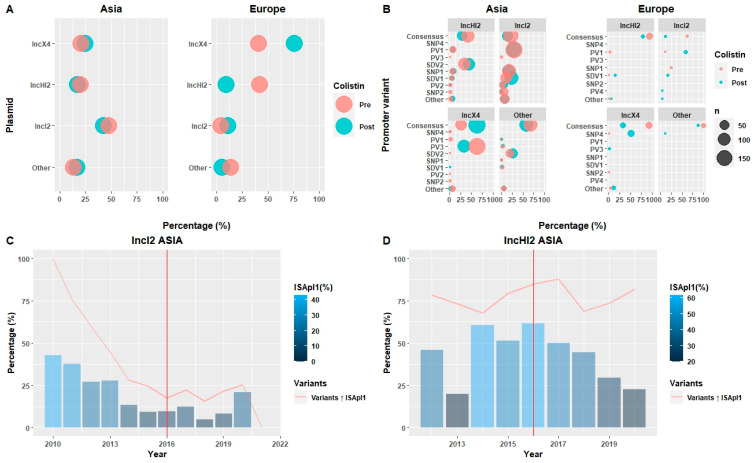
Distinct adaptation of *mcr-1* genomic context following colistin restriction in Asia and Europe. (**A**,**B**) promoter variant frequency comparison before and after the colistin ban in Asia and Europe. The size of the circle is proportional to the number of isolates of each category. (**C**) The percentage of IncI2 Asian isolates from 2010 to 2021 in which IS*Apl1* is conserved compared to the frequency of variants associated with IS*Apl1* (red line). (**D**) The percentage of IncHI2 isolates in Asia in which IS*Apl1* is conserved from 2012 to 2020 compared to the frequency of variants associated with IS*Apl1* (red line). The vertical red line marks the colistin restriction.

**Table 1 antibiotics-13-00759-t001:** Frequency of the distinct plasmid types and the carriage of other antimicrobial resistance genes.

Incompatibility Group	Number of Plasmids (%)	Mean Number of AMR Genes in the Plasmid	Mean Number of AMR Genes in the Whole Genome
IncI2	1231 (40.4)	1.3	13.3
IncX4	813 (26.7)	1.5	11.4
IncHI2	574 (18.8)	14.6	18.1
p0111	151 (5)	4	14.3
IncHI1	69 (2.3)	7	12
IncP1	62 (2.0)	1.7	11.3
IncB/O/K/Z	9 (0.3)	3.4	9.7
IncN	8 (0.3)	10.	15.5
IncFIA(HII)	8 (0.3)	7.4	15.2
IncX3	5 (0.2)	2.8	21.4
Not Determined	102 (3.34)	4.9	12.4

## Data Availability

Data are summarised in Appendix A. All assemblies are available in NCBI Pathogen Detection (https://www.ncbi.nlm.nih.gov/pathogens/), and raw data can be consulted in Appendix A.

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
