# Peer review of "Global Variation in Escherichia coli mcr-1 Genes and Plasmids from Animal and Human Genomes Following Colistin Usage Restrictions in Livestock"

_antibiotics, 2024, doi:10.3390/antibiotics13080759_

Round 1

Reviewer 1 Report

Comments and Suggestions for Authors

 Comments and Suggestions for Authors

As the first point, the title requires correction: the scientific name of the bacterial species and the gene name should be italicized for proper formatting.This error occurs numerous times throughout the manuscript with E. coli, Escherichia coli, mcr-1. The same applies to Isapl1 throughout the manuscript, and to Paenibacillus polymyxa on line 42, Enterobacteriaceae on line 57, and Actinobacillus pleuropneumoniae on line 66.

On line 12, a comma should be placed after the e-mail of B.P.

On line 42, the nomenclature of the subspecies is incorrect, as the subspecies name should begin with a lowercase letter and the correct spelling is colistinus.

On line 49, the authors incorrectly state that the gene mcr-1 was discovered in 2016. In fact, the gene was discovered in 2015, and the publication date is November 18, 2015. This imprecision is also present in the discussion section on line 235.

On line 56, it is suggested to change "not surprising" to "expected" to enhance the academic tone and formality of the manuscript.

On line 59, it is suggested to revise "without decreasing resistance rates" to "while maintaining resistance rates" for improved clarity and structure.

Please verify the accuracy of the information presented in lines 69-71. Numerous studies have extensively investigated the presence of this gene in animals, particularly in swine, poultry, rabbits, and cattle. This includes various European studies, specifically in Spain, Portugal, France, the United Kingdom, and Belgium. The entire spectrum of food-producing animals has been thoroughly examined within the One Health context.

On line 91, please specify the percentage threshold at which a result is considered notable. I suggest removing this subjective classification of results, as it introduces bias for the reader. If emphasis is desired for this result, it should be addressed in the discussion section.

On line 95, the phrase repeats "each isolate" twice. It is recommended to revise the sentence to: "... in silico, revealing an average of 13.6 AMR genes per isolate."

Please ensure that the formatting of the figure legends is consistent, including the font type and the use of italics. The legends should adhere to the specified formatting guidelines to maintain uniformity throughout the manuscript.

The information presented in lines 114-117 should be addressed in the Discussion section. The authors have highlighted this result but have not provided a discussion or interpretation of it. It is recommended to include this analysis in the appropriate section to provide a comprehensive understanding of the findings.

On line 116, please remove the parentheses following "hosts".

On line 134, please change "promotor" to "promoter." This correction is also needed in lines 137, 142, 205, and 333.

The information presented in line 135 should also be addressed in the Discussion section.

On lines 273, 330 and 332, the use of "et al." is incorrect. It should be revised to follow the appropriate citation style. Typically, "et al." should be italicized and followed by a period, like this: "Smith et al." Ensure the format complies with the specific style guide being followed by MDPI journals.

The statement in the text that "promoter variants related with a reduced fitness cost and ISApl1 showed distinct pattern of association that were responsible for the adaptation to colistin restriction" is strong and may be speculative without robust experimental data demonstrating direct causality. This information appears in the abstract and should be suggested rather than stated categorically in the discussion. A more cautious phrasing, such as "appear to be associated with," would be more appropriate.

The interpretation that the decrease in ISApl1 in Asia is due to changes in the prevalence of promoter variants could be better substantiated. Correlation does not imply causation, and other factors may be involved. It is important to discuss alternative potential causes and acknowledge the study's limitations in this context.

It is important to question whether a statistical analysis was conducted. If so, please specify the tests used. The results and conclusions would be considerably more robust and supported if correlation tests were performed. For example, a Chi-square (χ²) test could assess the association between the presence of different promoter variants and geographical regions (continents), determining whether the variants are evenly distributed or significantly associated with specific regions. Additionally, ANOVA could be used to compare the mean frequencies of promoter variants across different regions or hosts, identifying statistically significant differences among groups if more than two are involved. Please note that this is a suggestion for future research rather than a request for inclusion of such analyses in the current manuscript. Implementing these analyses in future studies could greatly enhance the robustness and significance of the findings.

Finally, it should be noted that there are inconsistencies in the formatting of the references. Terms such as mcr-1, Enterobacteriaceae, Pseudomonas aeruginosa, Escherichia coli, and ISApl1 should be italicized. Additionally, although mcr-1 is included in the title, it should be written in lowercase throughout the manuscript.

In references 7, 32, 38, 39, and 45, the DOI is missing. Please include the DOI for these references to ensure complete and accurate citation.

In the references, all entries have the year in bold except for references 3, 7, 29, 32, 38, 39, 45, and 47. Please ensure that the formatting is consistent by bolding the year for these references as well.

Reference 29 incorrectly ends with a semicolon (;). Please correct this by removing the semicolon and add a dot (.)

Given that the topic of colistin and mcr-1 is highly current and the subject of numerous scientific publications in various international journals, I expected that the majority of references would be from 2020 onwards. However, only 21 out of 49 references are from 2020 or later, representing less than 50% of the total. A quick search on PubMed using the keywords "mcr-1," "colistin," and "fitness cost" reveals several recent entries from 2020 onward, many of which reflect the European context. It is recommended to include more recent references to reflect the latest research developments and ensure that the manuscript is up-to-date.

Comments on the Quality of English Language

Yes, there are some minor issues related to the language, which have been addressed in the suggestions to the authors.

Author Response

Dear reviewer,

We would like to thank you for your time and the help that you provided us. Your feedback is highly appreciated and your comments have been very helpful.

Reviewer: As the first point, the title requires correction: the scientific name of the bacterial species and the gene name should be italicized for proper formatting.This error occurs numerous times throughout the manuscript with E. coli, Escherichia coli, mcr-1. The same applies to Isapl1 throughout the manuscript, and to Paenibacillus polymyxa on line 42, Enterobacteriaceae on line 57, and Actinobacillus pleuropneumoniae on line 66.

Authors: Thank you very much for detecting these typos. They have been corrected across the manuscript.

Reviewer: On line 12, a comma should be placed after the e-mail of B.P.

Authors: Many thanks for notice it. We have added.

Reviewer: On line 42, the nomenclature of the subspecies is incorrect, as the subspecies name should begin with a lowercase letter and the correct spelling is colistinus.

Authors: Thanks for pointing that out. Now it is solved.

Reviewer: On line 49, the authors incorrectly state that the gene mcr-1 was discovered in 2016. In fact, the gene was discovered in 2015, and the publication date is November 18, 2015. This imprecision is also present in the discussion section on line 235.

Authors: We are grateful for your observation. We only looked at the date of the citation, but the publication date is clear. We have changed! Many thanks for the correction!

Reviewer: On line 56, it is suggested to change "not surprising" to "expected" to enhance the academic tone and formality of the manuscript.

Authors: We have adapted the suggestion

Reviewer: On line 59, it is suggested to revise "without decreasing resistance rates" to "while maintaining resistance rates" for improved clarity and structure.

Authors: Thanks! The change is implemented in the manuscript!

Reviewer: Please verify the accuracy of the information presented in lines 69-71. Numerous studies have extensively investigated the presence of this gene in animals, particularly in swine, poultry, rabbits, and cattle. This includes various European studies, specifically in Spain, Portugal, France, the United Kingdom, and Belgium. The entire spectrum of food-producing animals has been thoroughly examined within the One Health context.

Authors: Here, the idea that we wanted to transmit was another one. We agree that mcr-1 has been widely studied in livestock, companion animals, wildlife or environment. The thing that we want to express here is that studies analysing the impact of colistin restriction have been done only in China and only one was including livestock samples. To make it clearer we have changed the order of the text: “Furthermore, despite evidence of regional differences in mcr-1 plasmid epidemiology [23] studies following recent restriction on colistin use have been confined to isolates of Chinese origin [24]. However, thesestudies have typically only investigated isolates from human infections [19,21], and it is important to include isolates from agricultural livestock animals, as a likely gene pool reservoir [25].”

Reviewer: On line 91, please specify the percentage threshold at which a result is considered notable. I suggest removing this subjective classification of results, as it introduces bias for the reader. If emphasis is desired for this result, it should be addressed in the discussion section.

Authors: We agree with this and we have removed the word notable: “Europe also contributed 11.3% of the samples (n=359)”

Reviewer: On line 95, the phrase repeats "each isolate" twice. It is recommended to revise the sentence to: "... in silico, revealing an average of 13.6 AMR genes per isolate."

Authors: Thanks for your suggestion! We have incorporated it in the text

Reviewer: Please ensure that the formatting of the figure legends is consistent, including the font type and the use of italics. The legends should adhere to the specified formatting guidelines to maintain uniformity throughout the manuscript.

Authors: Thanks for the comment. We have reviewed all the formatting mistakes presented and we have uniformed it.

Reviewer: The information presented in lines 114-117 should be addressed in the Discussion section. The authors have highlighted this result but have not provided a discussion or interpretation of it. It is recommended to include this analysis in the appropriate section to provide a comprehensive understanding of the findings.

Authors:  Thank you for the comment! Here, we opted for not including it in discussion, since we consider it a minor result, because, even the percentage of isolates with chromosome location was higher than in the other species, 90 % of the swine isolates were still located in the plasmids. Moreover, our explanation of this finding was very speculative based on the higher presence of ISApl1 sequences of the swine isolates. However, we opted for finally not presenting this data since was adding complexity to our already long manuscript. Then, discussing this (in our opinion, minor) finding was difficult and we obviate it. Personally, we think that it is better to not modify this point. Nevertheless, if it is required, we could simply delete this from the results since it is not a key finding.

Reviewer: On line 116, please remove the parentheses following "hosts".

Authors: Thanks for detecting it. We have removed it!

Reviewer: On line 134, please change "promotor" to "promoter." This correction is also needed in lines 137, 142, 205, and 333.

Authors: Many thanks for noticing the typo! All the corrections are done!

Reviewer: The information presented in line 135 should also be addressed in the Discussion section.

Authors: Thanks for the suggestion! We have added this text in the discussion: “The colistin ban has led to a decrease in ISApl1 [21,24] and the emergence of mutations in the mcr-1 promoter region [16], reducing its fitness cost. We observed a relationship between both phenomena.  ISApl1 presence was rare in IncX4 plasmids as previously described [19,36]. For IncI2, two trends were observed: promoter variants associated with the presence of ISApl1 and those without it. The most common variant, PV1, does not confer a fitness advantage on its own, explaining its low association with ISApl1, which could otherwise lead to plasmid destruction [27]. SDV1, another common variant, is linked to the loss of ISApl1 in IncI2 plasmids and provides a fitness benefit. This is in contrast to its increased presence of ISApl1 in IncHI2 plasmids. Further studies are needed to fully understand the relationship between plasmid type, promoter variants, and ISApl1 presence to comprehend mcr-1 adaptations after the colistin ban, since it seems that this relationship could vary according the distincts plasmids.”  This aspect is still difficult to explain and it is not easy to make it simple, but we hope that is clear for the reader.

Reviewer: On lines 273, 330 and 332, the use of "et al." is incorrect. It should be revised to follow the appropriate citation style. Typically, "et al." should be italicized and followed by a period, like this: "Smith et al." Ensure the format complies with the specific style guide being followed by MDPI journals.

Authors: We want to empathise this concrete work in these cases. We have changed for “and colleagues”. However, if the editor or the reviewers consider that is not adequate, we will change it to other format.

Reviewer: The statement in the text that "promoter variants related with a reduced fitness cost and ISApl1 showed distinct pattern of association that were responsible for the adaptation to colistin restriction" is strong and may be speculative without robust experimental data demonstrating direct causality. This information appears in the abstract and should be suggested rather than stated categorically in the discussion. A more cautious phrasing, such as "appear to be associated with," would be more appropriate.

Authors: We appreciated your suggestion and we think that is more appropriate. We have changed the expression.

Reviewer: The interpretation that the decrease in ISApl1 in Asia is due to changes in the prevalence of promoter variants could be better substantiated. Correlation does not imply causation, and other factors may be involved. It is important to discuss alternative potential causes and acknowledge the study's limitations in this context.

Authors: Given the large number of variables that can affect this, we have softened the language across the text to make clearer that is an association. Moreover, we suggest in the discussion with a sentence that more research is needed to understand this association and other factors since increased conjugation could be the responsible: “This relationship is difficult to understand, since it is not clear how a single nucleotide mutation could be associated with the loss of an insertion sequence and requires further investigation, since other factors such increase of plasmid transmission could be implicated”.

Reviewer: It is important to question whether a statistical analysis was conducted. If so, please specify the tests used. The results and conclusions would be considerably more robust and supported if correlation tests were performed. For example, a Chi-square (χ²) test could assess the association between the presence of different promoter variants and geographical regions (continents), determining whether the variants are evenly distributed or significantly associated with specific regions. Additionally, ANOVA could be used to compare the mean frequencies of promoter variants across different regions or hosts, identifying statistically significant differences among groups if more than two are involved. Please note that this is a suggestion for future research rather than a request for inclusion of such analyses in the current manuscript. Implementing these analyses in future studies could greatly enhance the robustness and significance of the findings.

Authors: Thank you for the suggestion. We use chi-square tests across the text and were highly significant. Now we have added to the text. Regarding ANOVA, it would be nice to incorporate in future studies. Thanks for the idea!

Reviewer: Finally, it should be noted that there are inconsistencies in the formatting of the references. Terms such as mcr-1, Enterobacteriaceae, Pseudomonas aeruginosa, Escherichia coli, and ISApl1 should be italicized. Additionally, although mcr-1 is included in the title, it should be written in lowercase throughout the manuscript. In references 7, 32, 38, 39, and 45, the DOI is missing. Please include the DOI for these references to ensure complete and accurate citation. In the references, all entries have the year in bold except for references 3, 7, 29, 32, 38, 39, 45, and 47. Please ensure that the formatting is consistent by bolding the year for these references as well.Reference 29 incorrectly ends with a semicolon (;). Please correct this by removing the semicolon and add a dot (.)

Authors: Thanks for notice the problems with the references. We experimented some problems with our reference manager, so we have manually modified. Now references are in the correct format.

Reviewer: Given that the topic of colistin and mcr-1 is highly current and the subject of numerous scientific publications in various international journals, I expected that the majority of references would be from 2020 onwards. However, only 21 out of 49 references are from 2020 or later, representing less than 50% of the total. A quick search on PubMed using the keywords "mcr-1," "colistin," and "fitness cost" reveals several recent entries from 2020 onward, many of which reflect the European context. It is recommended to include more recent references to reflect the latest research developments and ensure that the manuscript is up-to-date.

Authors: We appreciate your suggestion. Thus, we have included 7 new references from 2022 onward.

Reviewer 2 Report

Comments and Suggestions for Authors

Title of reviewed article:

Global variation in Escherichia coli mcr-1 genes and plasmids from animal and human genomes following colistin usage restrictions in livestock

Comments

Relevance of the title

There is relevance of title and the results.

Abstract

The aim of the study is not immediately clear. In the first lines, the authors stated the problem of AMR and the use of colistin. Then the authors state the findings “…This study analyzes 3163 Escherichia coli genomes with the mcr-1 gene from human and livestock hosts, mainly from Asia (n=2621) and Europe (n=359)” (lines 18 to29). It will be good if authors can clarify in a sentence or two in the abstract their aim of the study? Is it through policy changes that ban colistin use in China and Europe, the circumstance creates a natural experiment or condition to study the genetic background of mcr-1. The condition or reduction by policy can help study the different transmission routes, differences, and similarities of resistant genes in various geographic locations. It will be clearer study aim if the authors can state their goal(s).

Alignment of aim and findings of the study

The study of mcr-1 gene and ban of colistin is a significant study. This study reviewed mcr-1 gene in various countries. The alignment becomes clear as the reader goes through the finding section but not available especially in the abstract.

Authors please kindly use causation  terminology such as affect carefully. There is usually a great deal of caution in calling an association a causation, especially between colistin use in farm and mcr-1 abundance in human. Authors in this study describe “global agricultural livestock networks” have affected geographical differences in the prevalence of mcr-1 plasmid types in human clinical cases. Authors suggested “… (we) will investigate how global agricultural livestock networks have affected geographical differences in the prevalence mcr-1 plasmid types in human clinical cases” page 4 lines 143 and 151. Many researchers and investigators have established some form of association on this topic, but not causation. If there is a reason the authors of this study can establish a causal relationship, please state with clear reasons such as establishing a data-based epidemiological pathway(s), or a statistical approach that establishes causality. If not, perhaps it will be a good practice to state an association and not a causation.

Definition and clarification

Can authors please define their “One Health approach” and “global agricultural livestock networks”? If authors are taking human and livestock sectors into consideration, what about the environmental sector such as soil and water, and ecosystem which is defined as One Health framework studying AMR eg World Health Organization (www.who.int)?

Can authors please state rationale in choice of timeline? The authors are correct in stating there is a colistin use reduction in 2017. Please clarify why further timeline such as 2020 July’s implementation to ban colistin as a result of “The Chinese National Action Plan to Combat Antimicrobial Resistance (2017–2020) launched to eliminate the use of antibiotics in livestock feed by 2020”. Or the earlier than 2017 timeline for The European Medicines Agency to implement the policy to withdraw colistin by 2016.

Findings

Section 2.3 first paragraph page 4 lines 130 to 135 reads as introduction material and not findings. Please relocate this writing.

Minor changes

Missing noun “we” in line 77 before “will” or typo please change “will” to “we”.

For the percentages, would it be possible to provide (nominator/denominator) or n=number for lines 139 to 141 on page 4? Same for page 5 lines 163 to 164 and 171 to 179.

The spacings between the paragraphs throughout the writing is inconsistent, please change them accordingly.

Please use numerical numbers or English writing in numbers consistently. Please check and change throughout the writing. Is there a reason for writing 21 as twenty-one and 16 as 16?

Method and material

Please rewrite in Method and Material as some of the sentences that are dangling sentences for example “…Apart from the 16 already described promoter variants, we detected twenty-one more” lines 344 to 345 for example “…We detected 21 promoter variants in addition to  16 (core/primary/previously identified) variants”. Please use numerical numbers and English writing in numbers consistently.

Comments on the Quality of English Language

Some minor typo and dangling sentences. 

Author Response

Dear reviewer,

We are grateful about your contribution, and we think that your revision has improved our manuscript. Your feedback has been highly appreciated.

Reviewer: Abstract

The aim of the study is not immediately clear. In the first lines, the authors stated the problem of AMR and the use of colistin. Then the authors state the findings “…This study analyzes 3163 Escherichia coli genomes with the mcr-1 gene from human and livestock hosts, mainly from Asia (n=2621) and Europe (n=359)” (lines 18 to29). It will be good if authors can clarify in a sentence or two in the abstract their aim of the study? Is it through policy changes that ban colistin use in China and Europe, the circumstance creates a natural experiment or condition to study the genetic background of mcr-1. The condition or reduction by policy can help study the different transmission routes, differences, and similarities of resistant genes in various geographic locations. It will be clearer study aim if the authors can state their goal(s).

Alignment of aim and findings of the study

The study of mcr-1 gene and ban of colistin is a significant study. This study reviewed mcr-1 gene in various countries. The alignment becomes clear as the reader goes through the finding section but not available especially in the abstract.

Author: We appreciate for your comment. We have added in the abstract the following sentences: “In consequence, its livestock use was banned in 2017, originating a natural experiment to study the bacterial adaptation. The aim of this work was to analyze the changes in the mcr-1 genetic background after colistin restriction across the world.” We hope that now the aim of the study is clearer.

Reviewer: Authors please kindly use causation  terminology such as affect carefully. There is usually a great deal of caution in calling an association a causation, especially between colistin use in farm and mcr-1 abundance in human. Authors in this study describe “global agricultural livestock networks” have affected geographical differences in the prevalence of mcr-1 plasmid types in human clinical cases. Authors suggested “… (we) will investigate how global agricultural livestock networks have affected geographical differences in the prevalence mcr-1 plasmid types in human clinical cases” page 4 lines 143 and 151. Many researchers and investigators have established some form of association on this topic, but not causation. If there is a reason the authors of this study can establish a causal relationship, please state with clear reasons such as establishing a data-based epidemiological pathway(s), or a statistical approach that establishes causality. If not, perhaps it will be a good practice to state an association and not a causation.

Author: We appreciate this comment. We think that maybe we were too assertive referring causation. Thus, we have softened the language across the manuscript to indicate association.

Reviewer: Definition and clarification

Can authors please define their “One Health approach” and “global agricultural livestock networks”? If authors are taking human and livestock sectors into consideration, what about the environmental sector such as soil and water, and ecosystem which is defined as One Health framework studying AMR eg World Health Organization (www.who.int)?

Author: Our initial idea was to include genomes from environment. However, when we created the collection, there was less than 100 genomes available and from different origins (wild animals, water, plants), difficulting to merge it in only one category. For this reason, we opted for not including them. To avoid confusion, we have deleted the sentence in the introduction.

Reviewer: Can authors please state rationale in choice of timeline? The authors are correct in stating there is a colistin use reduction in 2017. Please clarify why further timeline such as 2020 July’s implementation to ban colistin as a result of “The Chinese National Action Plan to Combat Antimicrobial Resistance (2017–2020) launched to eliminate the use of antibiotics in livestock feed by 2020”. Or the earlier than 2017 timeline for The European Medicines Agency to implement the policy to withdraw colistin by 2016.

Author: Thank you for pointing this. The aim of the study was to analyse the mcr-1 adaptation to the cessation of colistin use. It is true that, in Europe, the legislation was approved in 2016. However, the colistin sales data shows that the reduction occurred in 2017 (European Medicines Agency, 2019), so this date was a better choice. This consideration allowed us to use the same date that in China (Walsh & Wu, 2016), that was already used in the previous studies comparing the situation before and after restriction there (Jiang et al., 2020; Ogunlana et al., 2023; C. Shen et al., 2020; Y. Shen et al., 2022). However, this cessation of use not evenly applied is a possible limitation of the study. In consequence, we stated in the discussion: “Furthermore, as the colistin ‘ban’ has not been applied evenly between countries, from complete prohibition to cessation of prophylactic and growth-promoting but continued therapeutic use under strict conditions”. We hope that your concern is solved.

Reviewer: Findings

Section 2.3 first paragraph page 4 lines 130 to 135 reads as introduction material and not findings. Please relocate this writing.

Author: Thanks for your suggestion. Since we already explained this in the introduction and, here, we were repeating it to reinforce the message, we have opted for eliminating it.

Reviewer: Minor changes

Missing noun “we” in line 77 before “will” or typo please change “will” to “we”.

Author: Thanks for noticing this! We have corrected the typo.

Reviewer: For the percentages, would it be possible to provide (nominator/denominator) or n=number for lines 139 to 141 on page 4? Same for page 5 lines 163 to 164 and 171 to 179.

Author: Thank you for pointing out this. We have incorporated in the manuscript

Reviewer: The spacings between the paragraphs throughout the writing is inconsistent, please change them accordingly.

Author: Thanks for detecting these mistakes. We have solved it!

Reviewer: Please use numerical numbers or English writing in numbers consistently. Please check and change throughout the writing. Is there a reason for writing 21 as twenty-one and 16 as 16?

Author: We appreciate your comment. We have changed in the text!

Reviewer: Method and material

Please rewrite in Method and Material as some of the sentences that are dangling sentences for example “…Apart from the 16 already described promoter variants, we detected twenty-one more” lines 344 to 345 for example “…We detected 21 promoter variants in addition to  16 (core/primary/previously identified) variants”. Please use numerical numbers and English writing in numbers consistently.

Author: We have revised the section to improve the redaction. We hope that now is clearer.

Bibliography:

European Medicines Agency. (2019). Sales of veterinary antimicrobial agents in 31 European countries in 2018: Trends from 2010-2018. Tenth ESVAC Report - EMA.

Jiang, Y., Zhang, Y., Lu, J., Wang, Q., Cui, Y., Wang, Y., Quan, J., Zhao, D., Du, X., Liu, H., Li, X., Wu, X., Hua, X., Feng, Y., & Yu, Y. (2020). Clinical relevance and plasmid dynamics of mcr-1-positive Escherichia coli in China: a multicentre case-control and molecular epidemiological study. The Lancet Microbe, 1(1). https://doi.org/10.1016/S2666-5247(20)30001-X

Ogunlana, L., Kaur, D., Shaw, L. P., Jangir, P., Walsh, T., Uphoff, S., & MacLean, R. C. (2023). Regulatory fine-tuning of mcr-1 increases bacterial fitness and stabilises antibiotic resistance in agricultural settings. The ISME Journal. https://doi.org/10.1038/s41396-023-01509-7

Shen, C., Zhong, L. L., Yang, Y., Doi, Y., Paterson, D. L., Stoesser, N., Ma, F., El-Sayed Ahmed, M. A. E. G., Feng, S., Huang, S., Li, H. Y., Huang, X., Wen, X., Zhao, Z., Lin, M., Chen, G., Liang, W., Liang, Y., Xia, Y., … Tian, G. B. (2020). Dynamics of mcr-1 prevalence and mcr-1-positive Escherichia coli after the cessation of colistin use as a feed additive for animals in China: a prospective cross-sectional and whole genome sequencing-based molecular epidemiological study. The Lancet Microbe, 1(1). https://doi.org/10.1016/S2666-5247(20)30005-7

Shen, Y., Zhang, R., Shao, D., Yang, L., Lu, J., Liu, C., Wang, X., Jiang, J., Wang, B., Wu, C., Parkhill, J., Wang, Y., Walsh, T. R., Gao, G. F., & Shen, Z. (2022). Genomic Shift in Population Dynamics of mcr-1-positive Escherichia coli in Human Carriage. Genomics, Proteomics & Bioinformatics. https://doi.org/10.1016/j.gpb.2022.11.006

Walsh, T. R., & Wu, Y. (2016). China bans colistin as a feed additive for animals. In The Lancet Infectious Diseases (Vol. 16, Issue 10). https://doi.org/10.1016/S1473-3099(16)30329-2